# “What Are the Applications for Remote Rehabilitation Management in Cystic Fibrosis?”: A Scoping Review Protocol

**DOI:** 10.3390/ijerph192114014

**Published:** 2022-10-27

**Authors:** Matteo De Marchis, Matteo Cioeta, Mario Cannataro

**Affiliations:** 1Department of Paediatrics Specialistics, IRCCS Bambino Gesù Paediatric Children’s Hospital, Piazza di Sant’Onofrio 4, 00165 Rome, Italy; 2Department of Neurological and Rehabilitation Sciences, IRCCS San Raffaele Roma, Via della Pisana 235, 00163 Rome, Italy; 3Department of Medical and Surgical Sciences, University “Magna Graecia” of Catanzaro, Viale Europa, 88100 Catanzaro, Italy

**Keywords:** cystic fibrosis, telemedicine, telehealth, scoping review, protocol

## Abstract

Background: Telemedicine is an effective, widely used strategy in the field of cystic fibrosis management. The objective of this scoping review is to summarize and analyze the scientific literature with the special focus on the tools and the strategies used in patients with a chronic disease, such as cystic fibrosis. Methods: This scoping review will be performed in accordance with the Joanna Briggs Institute methodology. In this context, the planned scoping review is a research synthesis that will map the literature on the applications of telemedicine and telemonitoring to the management of cystic fibrosis, with the aim to identify key concepts in the research and work to be conducted that may impact clinical practice. Studies will be included if they meet the following population, concept, and context criteria: all patients with cystic fibrosis receiving treatment with the tools of telemedicine and telemonitoring. No study design, publication type, or data restrictions will be applied. MEDLINE, Scopus, CINHAL, Pedro, Embase, Web of Science, ACM Digital Library, Health Technology Assessment Database (HTA), and Cochrane Central will be searched up to September 2022. Discussion: To the best of our knowledge, this will be the first scoping review to provide a comprehensive overview of the topic. The results could add meaningful information for future research and, especially, for clinical practice, when implementing telerehabilitation in cystic fibrosis treatment. Furthermore, we expect that our work may identify possible knowledge gaps on the topic. The results of this research will be published in a peer-reviewed journal and will be presented at relevant international scientific events, such as in congress or meetings.

## 1. Introduction

Cystic fibrosis (CF) is a progressive, multisystem disease that requires lifetime treatment [1]. CF is the most common life-limiting autosomal recessive disease in Caucasian populations, affecting approximately 5000 people in Italy [2]. Most people with CF will require lifelong treatment involving frequent hospital visits and admissions and rigorous daily therapy regimens (oral treatment, aerosol therapy, airway cleaning, and also physical activity rehabilitation). In particular, rehabilitation therapies can include the performance of chest physiotherapy (airway clearance) and exercise; the monitoring of oxygen; and the application of noninvasive ventilation [3]. The average daily time associated with treatment has been estimated at over 1.5 h [4]. This high level of treatment burden has a substantial impact on health-related quality of life (HRQoL), and is associated with reduced adherence [5,6,7]. Since many of these treatments necessitate frequent hospital visits for assessment and ongoing management, telemedicine may be useful. Telemedicine is defined as a direct, synchronous, or remote communication between a physician and a patient [8]. It is an emerging area of study in several chronic diseases, especially in CF patients. The pandemic has substantially increased the use of tools for remote monitoring for these patients: of 74 titles recovered in PubMed (Dec 2020) with the term “cystic fibrosis” and “telemedicine or telehealth”, 30% of them were published in 2020 [9]; most of the US and the UK CF clinics, therefore, went completely virtual, implementing systems to support remote care [10,11]. There is also growing acceptance among patients that they can be followed remotely during rehabilitation [12]. Despite this growing interest in the subject, the last systematic revision about this topic dates back to 2012 and it concluded that there is insufficient evidence to reach a firm conclusion about the benefits of telehealth in people with CF [13]. An update is necessary, but even before a classification as rigorous as possible regarding the types of telerehabilitation platforms developed in recent years, it needs to be acknowledged that currently, there are a lot of digital instruments or tools to improve the use of telecare, but there is not a map to classify and understand all the applications for remote control in cystic fibrosis. This heterogeneity of instruments could be considered, and it is essential to analyze all instruments more in detail.

Therefore, the objective of this project is to summarize the literature to identify the applications for remote control in cystic fibrosis.

As maintained by the Joanna Briggs Institute (JBI) [14], a scoping review approach may be used to map and clarify key concepts, identify gaps in the research knowledge base, and report the types of evidence that address and inform practice in the field. Scoping reviews are helpful when the literature is complex and heterogeneous [15,16].

In particular, the objectives of this study will be to:Provide a comprehensive overview of all studies dealing with applications in cystic fibrosis for remote rehabilitation;Identify a map to track digital methodologies on the topic;Share review findings with the scientific community.

## 2. Materials and Methods

The proposed scoping review will be conducted in accordance with the Joanna Briggs Institute methodology (JBI) for scoping reviews. The Preferred Reporting Items for Systematic Reviews and Meta-Analyses extension for Scoping Reviews (PRISMA-ScR) Checklist for reporting will be used [17].

### 2.1. Research Question

We formulated the following research question: “What are the applications for remote control in cystic fibrosis?”

The Population, Concept and Context (PCC) framework helped guide the formulation of the review question:-Population: this review will consider eligible studies that include all studies in which people with CF have used tools of telerehabilitation.-Concept: this review will consider eligible studies that include telerehabilitation as part of the intervention for patients with CF.-Context: this review will include studies conducted in any context.

### 2.2. Inclusion and Exclusion Criteria

This scoping review will consider any study designs or publication types for inclusion. No time, geographical, or setting restrictions will be applied. Studies published in the English or Italian language will be included. Studies that do not meet the above-stated Population-Concept-Context (PCC) criteria or that provide insufficient information will be excluded.

### 2.3. Search Strategy

An initial limited search of MEDLINE was undertaken to identify articles on the topic. The words contained in the titles and abstracts of the relevant articles and the index terms used to describe all applications and tools used to develop a full search strategy for MEDLINE (which shows the search strategy for each database) will be considered. The search strategy, including all identified keywords and index terms, will be adapted for use in MEDLINE, Scopus, CINHAL, PEDro, Embase, Web of Science, ACM Digital Library, Health Technology Assessment Database, and Cochrane Central (Table 1) will be searched up to September 2022. In addition, the grey literature (e.g., from Google Scholar, direct contact with experts in the field of cystic fibrosis, and rehabilitation) and the reference lists of all relevant studies will be searched. The complete search strategy used for this scoping review will be reported according to the PRISMA-S (extension for reporting literature searches in systematic reviews) [18].

### 2.4. Study Selection

Once the search strategy has been successfully completed, search results will be collected and imported into EndNote X7 (Clarivate Analytics, Philadelphia, PA, USA). Duplicates will be automatically removed before the file containing a set of unique records is made available to reviewers for further processing (i.e., study screening and selection). The selection process will consist of two levels of screening using Rayyan QCRI online software [19]: (1) a title and abstract and (2) a full-text screening. For both levels, two investigators (M.D.M. and M.C.) will screen the articles independently to determine if they meet the inclusion/exclusion criteria. If there are disagreements, they will be resolved through discussion or a third reviewer (M.C.). Reasons for the exclusion of each record will be recorded and reported in a separate file in the final scoping review. The entire selection process will be reported and presented in a PRISMA flow diagram [20].

### 2.5. Data Extraction

A draft extraction tool is provided (see Table 2), which illustrates different telerehabilitation tools. This form will be reviewed by the research team and pretested by all reviewers before implementation to ensure that the form captures the information accurately. Charting results is commonly an iterative process during scoping reviews; other data can be added to this form according to the subgroups that could emerge from the analysis of the studies included. Modifications will be detailed in the full scoping review.

### 2.6. Data Management

As a scoping review, the purpose of this study is to aggregate the findings and present an overview on the research rather than to evaluate the quality of the single studies. The results will be presented in two ways:Numerically. Data extraction will be summarized in tabular form. Further categories may be added if considered appropriate.Thematically. A descriptive analysis will be performed pertaining to themes and key concepts relevant to the research questions and according to subgroups that could emerge.

## 3. Discussion

We are expecting many heterogeneous results about the numerous tools of telemedicine and telerehabilitation. We try to remove all bias inside papers found to discuss the possible comparisons between selected studies. Many studies will underline the personalization of cure, a focal point needed for health care. We are also expecting different values of adherence for rehabilitation treatment between adults patients and pediatric patients. The modality of administration of telerehabilitation treatment could be different, regarding platform, app, etc. A possible consideration will be that not all instruments are devices with health certifications, or they had been patented. To the best of our knowledge this will be the first scoping review on this topic and could improve future treatments in this area.

## 4. Conclusions

According to us, this work will underline the necessary approaches to organize and define tools for the management in cystic fibrosis. It is fundamental to create a road map to describe the current situation in cystic fibrosis. Remote rehabilitation management became one of the most important areas in telemedicine, especially during the COVID-19 outbreak. Telerehabilitation will be essential and, for the future, it represents a life-key for cure-management in CF.

## Figures and Tables

**Table 1 ijerph-19-14014-t001:** Search strategy.

Databse	Research Strategy
MEDLINE	(Cystic Fibrosis [MeSH Terms] OR “Cystic Fibrosis”) AND (Telehealth OR Telemedicine OR Telemedicine [MeSH Terms] OR Teleconsultation [MeSH Terms] OR Teleconsultations [MeSH Terms] OR teleconsult* OR Telerehab* OR Telemonitoring*)
Embase	‘cystic fibrosis’ AND (‘telehealth’ OR ‘telemedicine’ OR teleconsult* OR telerehab* OR telemonitoring*)
Web of Science	‘cystic fibrosis’ AND (‘telehealth’ OR ‘telemedicine’ OR teleconsult* OR telerehab* OR telemonitoring*)
CINHAL	‘cystic fibrosis’ AND (‘telehealth’ OR ‘telemedicine’ OR teleconsult* OR telerehab* OR telemonitoring*)
ACM Digital Library	‘cystic fibrosis’ AND (‘telehealth’ OR ‘telemedicine’ OR teleconsult* OR telerehab* OR telemonitoring*)
Health Technology Assessment Database (HTA)	‘cystic fibrosis’ AND (‘telehealth’ OR ‘telemedicine’ OR teleconsult* OR telerehab* OR telemonitoring*)
Cochrane Central	‘cystic fibrosis’ AND (‘telehealth’ OR ‘telemedicine’ OR teleconsult* OR telerehab* OR telemonitoring*)

**Table 2 ijerph-19-14014-t002:** Different telerehabilitation tools and their characteristics.

Author, Year, Name	Study Design	Type of Rehabilitation (Motor Rehabilitation, Respiratory Rehabilitation, Cognitive Rehabilitation)	Delivery Method (Synchronous and Asynchronous)	Delivery Session (Associated and Individual)	Number of Sessions	Duration of Session	Pretraining Session (Yes/No)	Type of Tool Used	Aim of Tool

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
