# Peer review of "“What Are the Applications for Remote Rehabilitation Management in Cystic Fibrosis?”: A Scoping Review Protocol"

_ijerph, 2022, doi:10.3390/ijerph192114014_

Round 1

Author Response

Abstract

Point 1: Furthermore, any knowledge gaps on the topic will be identified.

Response 1: Thank you for your suggestion about methodology; in the abstract, line 26, we replaced with “Furtermore, we exspect that our work may identify possibile knowledge gaps on the topic”

Point 2 Line 25: The results will add

Response 2: We replaced by “ The results could add meaningful information for future researches

Point 3 Line 25: Replace expecially with especially

Response: 3: We replaced expecially by especially

Point 4 Line 26 : Furthermore any knowledgegaps in the topic will be identified …

Response 4: Furthermore we expect that our work may identify possible gaps on the topic

Point 5 line 27: Replace bepublished by be published

Response 5: We replaced bepublishe by be published

Point 6 line 28: Replace scientificevents by scientific events. In this line remove the words important meetings as well as “meeting research and leave it as like congress or meetings”.

Response 6: We replaced scientficevents by scientific events and we leave in this line congress or meetings.

Introduction:

Point 1 line 31-41: Beacause this paper has a physicians as a main audience my suggestion, and this only a suggestion is to remove the all explanation about what iscystic fibrosis.

Response 1: We simplify this section, and we eliminated some literature references.

Point 2 line 43: Replace aerosoltherapy by aerosol therapy

Response 2: We replace aersoltherapy by aerosol therapy

Point 3 Line 59: Replace subjectthe by subject

Response 3: We replaced subjectthe by subject the

Point 4 line 63: correct ther by there

Response 4: We corrected ther by there

Point 5 line 65: Replace it’s with it is

Response 5: We replace it’s with It is

Point 6 line 71: Correct reporton with report on

Response 6: We correct reporton with report on

Methodology

Point 1: Why did the authors not include google scholar as a database for your research? This might cover a high number of papers in the topic of interest.

Response 1: We agree with your observation and based on results of Haddaway et al. 2015 (“The Role of Google Scholar in Evidence Reviews and Its Applicability to Grey Literature Searching”) we added at Line 110 “In addition, also grey literature (e.g. Google scholar, direct contact with experts in the field of cystic fibrosis and rehabilitation) and the reference lists of all relevant studies will be searched.

Point 2: Line 98 - replace Italianlanguage by Italian language. I suggest for the authors to include in their search only papers in English since a lot of readers won’t be able to consult these manuscripts due to the language barriers.

Response 2:  Line 98 - We replace Italianlanguage by Italian language. Furthermore we believe that our italian language knowledge could be a strenght point to the scientific community for reviewing eventually italian papers found in the literature search.

Point 3: In table 01, search strategy, I suggest include the search words “chest” “physiotherapy”

Response 3: In the first submission of the paper we decided to not included these specific words to avoid excessively restriction in the search strategy. For example in the MEDLINE search strategy, applying the word “physiotherapy” reduces results from 159 to 22.

Discussion

Point 1: What kind of bias the authors are referring to?

Response 1: With this sentence we inform the scientific community that we’re not going to analyze every single study that we will include in the scoping review with a specific tool (e.g. ROBIN-I, ROBIN-E), but that following the JBI guidelines the PRISMA-SR checklist we will give a strenght strategy to conduct future studies with more clear informations regarding for example the tools charateristics (see table 2) used in telerehabiitaion trials.

Conclusion

Point 1: Line 156 – “It’s fundamental…” replace by “It is fundamental”

Response 1: Line 156 – We replace “It’s fundamental” by “It is fundamental”

Reviewer 2 Report

This study analyzes the use of remote rehabilitation management in cystic fibrosis, but much of the research analysis needs to be strengthened. I recommend rejecting this manuscript, and the following comments should be taken into consideration by the authors.

The specific comments are as follows:

1. Abbreviations should be avoided as much as possible in the abstract section, and a period should be added to the last paragraph.

2. It is recommended that the quality of the included literature be assessed to ensure the reliability of the literature data.

3. The discussion section does not yield any noteworthy points, and the entire article focuses more on the methodology and steps of the study, without yielding valuable information about the final data.

4. For the two forms of presentation proposed in the data management section of 2.6, they are not fully demonstrated in the text, and it is suggested to analyze and discuss them in the discussion section.

5. The conclusions drawn in the conclusion section are not given a specific explanation in the text . This manuscript as a whole lacks analysis of the data and does not provide detailed explanations and discussions based on the information and data obtained from the search.

6. It is recommended to add subgroup analysis and publication bias analysis.

7. It is suggested that Table 2 should list the different telerehabilitation tools and their characteristics for each of the literature screened in this paper, rather than just including the characteristics.

8. It is recommended that the obtained results be subjected to correlation analysis.

Author Response

Our work concerns the drafting of a protocol for a subsequent scoping review to be published in a scientific journal. For this reason below we attached our replies to the comments considering the fact that it is a protocol of a revision and for this reason some elements requested cannot be present.

Point 1: Abbreviations should be avoided as much as possible in the abstract section, and a period should be added to the last paragraph

Response 1: We agree with your suggestion and replaced in Line 18 “CF” with cystic fibrosis. In our opinion the abstract is complete with the last paragraph and further comments regarding future results of the scoping review couldn’t be added.

Point 2: It is recommended that the quality of the included literature be assessed to ensure the reliability of the literature data.

Response 2: Since it is a scoping review and analyzing a new topic within the literature we decided not to insert resistances regarding the design of the studies included.

This will lead to the drafting of all types of study for example (RCT, pilot study, cohort study, cross-sectional study etc.). To obtain the most appropriate methodology for conducting the review we will use the JBI Guidelines and Preferred Reporting Items for Systematic reviews and Meta-Analyses extension for Scoping Reviews (PRISMA-scr) Checklist.

Point 3: The discussion section does not yield any noteworthy points, and the entire article focuses more on the methodology and steps of the study, without yielding valuable information about the final data.

Response 3: As mentioned above, a protocol of a scoping review doesn’t provide informations about final results, but protocol scoping review’s aim is to define the appropriate methodology that the autors will follow in the review. Final results will be reported in the scoping review.

Point 4: For the two forms of presentation proposed in the data management section of 2.6, they are not fully demonstrated in the text, and it is suggested to analyze and discuss them in the discussion section.

Response 4: The two forms of presentation proposed are not fully demonstrated in the text because in the protocol we don’t have results about our research. This suggestion about the presentation of the results can be fundamental for us in the presentation of the results in the following paper.

Round 2

Reviewer 2 Report

This paper focuses on methodology research. Please highlight the innovation of this study compared to existing studies.

Author Response

Thank you for your reply. In the "background section" we point out the need to review all the tools avaible in letterature to summarize and classify them to improve future researches and treatments.

Also we add "To the best of our knowledge this will be the first scoping review on this topic and could improve future treatments in this area." at the line 149-150.